# YouCLIP: Advancing Multilingual Cross-Modal Learning with Efficient Training.

## Abstract

Since the advent of vision-language pretraining, the CLIP model has become a foundational model for many downstream tasks. However, most of the advanced CLIP models available today are trained primarily on English, making them poorly suited for other languages. This limits accessibility for countries where other languages are dominant. Given that training CLIP models requires vast amounts of GPU resources and data, which most countries lack due to the absence of companies on the scale of Google or OpenAI, this paper proposes an efficient and straightforward three-stage fine-tuning method, which allows for the conversion of the most powerful English CLIP model into models for other languages. In these three stages of training, the first stage focuses on aligning the embedding layer, followed by token fusion in the second stage, and finally contrastive learning fine-tuning in the third stage. Meanwhile, to improve data quality, we propose a translation filtering model to filter the data. In this work, we target Chinese as the language of interest and name the resulting model YouCLIP, which is currently the most powerful Chinese CLIP model, significantly outperforming previous models across all Chinese benchmarks. For example, YouCLIP improves the text-to-image Recall@1 score on the COCO-CN dataset from 63.4 to 73.1. Additionally, YouCLIP retains strong English capabilities, achieving a Top-1 accuracy of 76.9 on ImageNet. Despite these impressive results, YouCLIP requires the least amount of training resources compared to other Chinese CLIP models. All models and code for YouCLIP will be open-sourced.

## 1 Introduction

Since the introduction of Contrastive Language-Image Pre-training (CLIP) (Radford et al., 2021) by OpenAI in 2021, vision-language models have significantly advanced, becoming foundational for a range of cross-modal applications, including image-text retrieval, zero-shot classification, and generative models. A multitude of approaches (Zhai et al., 2023; Fang et al., 2024; Sun et al., 2023) have been proposed, achieving remarkable success across a variety of downstream tasks, thereby revolutionizing our understanding and capabilities in bridging visual and linguistic modalities.

However, most current CLIP models are developed primarily for English, making it difficult for countries with other native languages, such as Chinese, Japanese, Korean, and smaller languages in less developed regions, to utilize them effectively. In this context, developing high-performance CLIP models for these languages at a low cost becomes especially important, as not every country has big companies like Google or OpenAI, which possess vast amounts of GPUs and image-text data.

One of the simplest methods for developing CLIP models in other languages is to directly use image-text pairs from those languages to train a CLIP model from scratch. However, this approach is costly and yields limited results. For instance, CN-CLIP (Yang et al., 2022) was trained using 64 V100 GPUs and 200 million image-text pairs, yet its performance still lags behind English CLIP models, which are often trained on billions of data points. Some papers have proposed methods for distilling the text side of the model (Carlsson et al., 2022; Chen et al., 2023), but these approaches tend to be relatively basic, offering low performance and limited practical applicability.

In this paper, we propose a low-cost, high-efficiency method for building CLIP models in other languages, which leverages the strongest English CLIP model as a foundation and transforms the

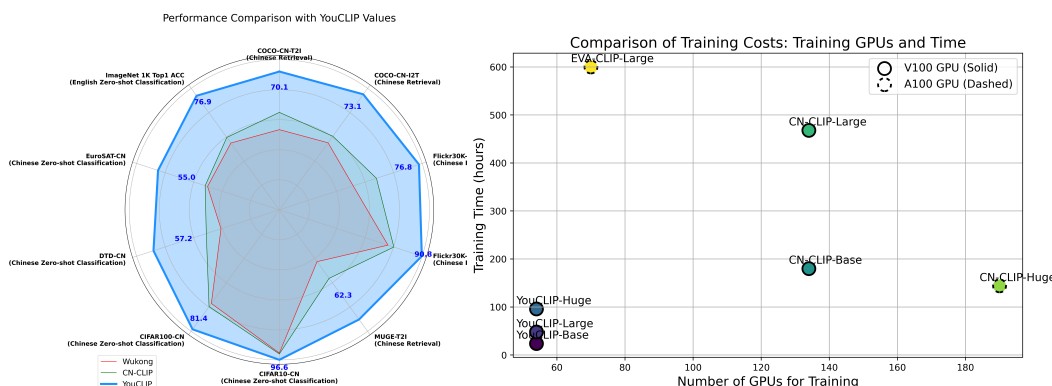

Figure 1: **Left**: Performance comparison of different Chinese Large-sized CLIP models. It can be seen that YouCLIP achieves the highest performance indicators in all tasks and datasets; **Right**: Comparison of the training costs of different models. It can be seen that the number of GPUs and training time used by YouCLIP are very small, and the overall training cost is very low.

English CLIP through a three-stage fine-tuning process, theoretically enabling the conversion of the top-performing English CLIP into a CLIP model for any target language. In this paper, we focus on Chinese as the target language and refer to the resulting CLIP model as YouCLIP, which is currently the most powerful Chinese CLIP model on all evaluation metrics and datasets as shown in Fig.(1).

To develop the strongest Chinese CLIP model, we approached the task from both data and methodology perspectives. **On the data side**, since most open-source datasets on the internet are in English, obtaining high-quality Chinese image-text pairs is relatively challenging. Therefore, we translated the captions in DFN2B (Fang et al., 2024) into Chinese using LLMs, resulting in approximately 1.5 billion English caption-Chinese caption-image triplets data. Additionally, we found that the outputs from the LLM can be unstable. To ensure the quality of the dataset, we developed a **Translation Filtering Network (TFN)** to filter out the highest quality 1.2 billion triplet data from the original 1.5 billion dataset for training.

**On the methodology side**, to reduce training costs and improve convergence speed, we proposed a novel **three-stage fine-tuning alignment strategy**. This strategy aligns the English text encoder with the Chinese encoder, followed by aligning the Chinese encoder with the image encoder. To begin with, let's break down the structure of CLIP. The image encoder is language-independent. The text side includes a tokenizer and a text encoder. The original English tokenizer cannot adapt well to Chinese tasks, so the tokenizer needs to be replaced. The text encoder can be broadly divided into an embedding layer and multiple Transformer layers. We observe that the primary difference of text encoders lies in the embedding layer. If the concepts in the embedding layer can be aligned, the subsequent Transformer layers can remain unchanged, resulting in identical outputs. Therefore, in the first stage, we first use the English encoder for supervision, reconstructing the vocabulary and training the embedding layer, while freezing the other parameters. The supervision comes from the proposed cross-language alignment, where the features of matched Chinese and English captions will be pulled closer together, while the features of unmatched captions will be pushed further apart. By using only text information for alignment, the need to download hundreds of terabytes of images is eliminated. On the other hand, since only a small portion of parameters needs to be trained, this accelerates both the training and convergence processes.

Since the same concept may be represented by different numbers of tokens in Chinese and English, perfect alignment is often unattainable. Therefore, we perform token fusion to solve the problem of inconsistent number of tokens for the same concept in different languages. The token fusion is performed with the first half of the Transformer layers of Chinese text Encoder under the supervision of cross-language alignment.

Due to the fact that the alignment loss values between the Chinese and English encoders never reach zero, there is always some gap between them. This results in an even larger gap between

the Chinese encoder and the inherited image encoder. Therefore, in the third phase, we directly employ the SigLIP (Zhai et al., 2023) loss to align the image encoder with the Chinese encoder. Since good alignment has already been achieved in the first two phases, we only use 150 million image-text pairs for minor fine-tuning in this step, significantly reducing training time and resource consumption. Additionally, to support both Chinese and English, we randomly sampled Chinese and English data from triplets during the three stages of training, which enabled YouCLIP to handle both languages effectively.

After three stages of fine-tuning, YouCLIP achieved highly satisfactory results. In both Chinese retrieval tasks and zero-shot classification, YouCLIP **significantly outperformed existing state-of-the-art models while maintaining a strong proficiency in English**, as shown in Fig.(1). YouCLIP is available in three different sizes: Base, Large, and Huge. For the Large model, the Recall@1 metric for Chinese text-to-image retrieval on the COCO-CN improved from the previous SOTA of **64.7 to 70.7**, while the image-to-text Recall@1 metric increased from **63.4 to 73.1**. Additionally, on the English ImageNet-1K dataset, YouCLIP improves the Top-1 accuracy from the previous state-of-the-art (SOTA) model's **59.6 to 76.9**. These results clearly demonstrate YouCLIP's robust performance.

## 2 RELATED WORK

### 2.1 VISION-LANGUAGE PRE-TRAINING

In the landscape of Vision-Language Pre-training (VLP), CLIP (Contrastive Language-Image Pre-training) (Radford et al., 2021) has emerged as a prominent dual-stream architecture, excelling at learning powerful cross-modal representations by separately encoding images and texts before aligning them in a joint space. While CLIP achieved notable success through large-scale pre-training on image-text pairs, subsequent research has sought to address its limitations and optimize its performance (Li et al., 2021; Gao et al., 2024; 2022; Fan et al., 2024). For example, EVA-CLIP (Sun et al., 2023) was trained on LAION-2B with 2 billion image-text pairs using 144 A100s. SigLiP (Zhai et al., 2023) was trained on the 10-billion-scale WebLi dataset using 32 TPUv4 units. DFN (Fang et al., 2024) filtered 10 billion image-text pairs to obtain 5 billion high-quality pairs for training.

It is evident that current CLIP models are becoming increasingly large in terms of data and computational requirements. Moreover, most CLIP models are designed specifically for English, while relatively few are tailored for other languages. This limits the use of CLIP models in non-English-speaking countries. The proposed YouCLIP in this paper, however, offers an efficient and convenient method to convert English CLIP models into other languages, such as Chinese.

### 2.2 NON-ENGLISH CLIP

Compared with English CLIP, there are relatively few CLIPs developed for non-English languages. There are generally two main approaches to developing these models: training from scratch and fine-tuning existing English CLIP models. For the first one, these approaches involves using non-English image-text pairs to build a CLIP model from the ground up. Classic examples of this method include CN-CLIP (Yang et al., 2022), WuKong (Gu et al., 2022), and Japanese-CLIP (Sawada et al., 2024). CN-CLIP is trained on 200 million Chinese image-text pairs, while WuKong utilizes 100 million Chinese image-text pairs. For the second approach, existing methods typically use techniques like Mean Squared Error (MSE) loss (Carlsson et al., 2022; Chen et al., 2023) to align the text encoders of Non-English and English. This enables the model to leverage the strengths of the pre-trained English CLIP while adapting it for Non-English through fine-tuning.

However, we must emphasize that existing methods are still insufficient. While training from scratch can effectively align images and Non-English text, it is often expensive and limited by the relatively small datasets used. On the other hand, fine-tuning based on English CLIP is more cost-efficient, but simply applying MSE loss results in poor alignment between Non-English text and images, leading to suboptimal performance. Our proposed YouCLIP addresses these issues with a three-stage fine-tuning process. Using a dataset of 1.2 billion image-text pairs, we have developed the most powerful Chinese CLIP model to date.

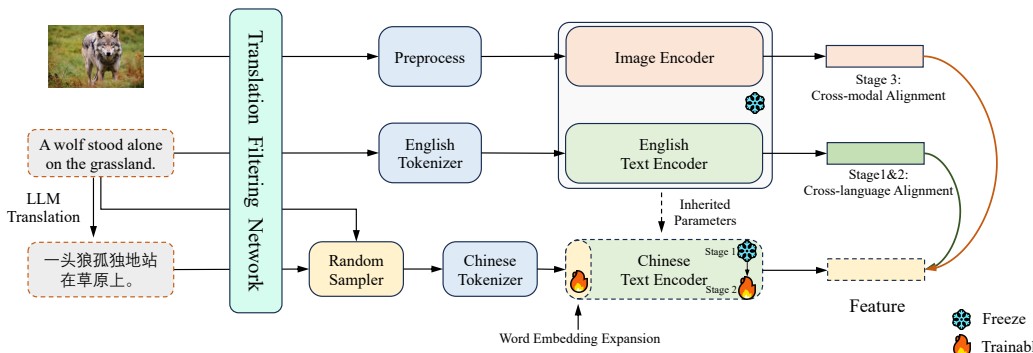

Figure 2: Framework of YouCLIP Training. First, the triplet data translated by the LLM is filtered through the Translation Filtering Network. Afterward, through three stages of fine-tuning driven by two sources of supervision signals, YouCLIP becomes the most powerful Chinese CLIP model.

## 3 METHOD

### 3.1 CLIP PRELIMINARIES

CLIP is designed to align visual and textual information through contrastive learning. Given a batch of $N_B$ image-text pairs $\{I_i, T_i\}_{i=1}^{N_B}$, the image $I_i$ and text $T_i$ are first processed by the image encoder $f_I(\cdot)$ and the text encoder $f_t(\cdot)$, respectively. The outputs are normalized with $L_2$ normalization $\phi(\cdot)$, resulting in the visual embeddings $\mathbf{z}_i^I$ and text embeddings $\mathbf{z}_i^T$:

$$\mathbf{z}_i^I = \phi\big(f_I(I_i)\big) \tag{1}$$

$$\mathbf{z}_i^T = \phi\big(f_t(T_i)\big) \tag{2}$$

These embeddings $\{(\mathbf{z}_i^I, \mathbf{z}_i^T)\}_{i=1}^{N_B}$ are used in contrastive learning through the InfoNCE loss to achieve cross-modal alignment. The goal is to bring corresponding image and text embeddings closer together in the embedding space, while pushing apart non-corresponding pairs. The following is a common form of infoNCE for training CLIP:

$$\mathcal{L}_{\text{InfoNCE}} = -\frac{1}{N_B} \sum_{i=1}^{N_B} \left( \log \frac{\exp(\mathbf{z}_i^T \cdot \mathbf{z}_i^I / \tau)}{\sum_{j=1}^{N_B} \exp(\mathbf{z}_i^T \cdot \mathbf{z}_j^I / \tau)} + \log \frac{\exp(\mathbf{z}_i^I \cdot \mathbf{z}_i^T / \tau)}{\sum_{j=1}^{N_B} \exp(\mathbf{z}_i^I \cdot \mathbf{z}_j^T / \tau)} \right) \tag{3}$$

where $\tau$ is the learnable temperature parameter. Based on this paradigm, many CLIP models have emerged, such as the DFN2B (Fang et al., 2024) and EVA-CLIP (Sun et al., 2023) for English, as well as CN-CLIP (Yang et al., 2022) and Wukong (Gu et al., 2022) for Chinese. These models leverage extensive data and computational resources, utilizing contrastive learning to align text and image modalities effectively.

### 3.2 FRAMEWORK OF YOUCLIP

Unlike high-cost, from-scratch training approaches like CN-CLIP (Yang et al., 2022), YouCLIP adopts a more efficient training method. It leverages the strongest English CLIP model as a teacher for distillation and inherits part of its parameters to achieve faster convergence, where the teacher is SigLIP models (Zhai et al., 2023) in the experiments.

Assuming the strongest English CLIP model contains an image encoder $f_I(\cdot)$ and and a text encoder $f_t^{\mathbf{E}}(\cdot)$, the goal of YouCLIP is to cost-effectively train a target language text encoder $f_t^{\mathbf{T}}(\cdot)$. This text encoder $f_t^{\mathbf{T}}(\cdot)$ can then pair with the original image encoder $f_I(\cdot)$ to form a CLIP model in the target language, where the target language is Chinese in our study.

Specifically, YouCLIP's training is divided into three stages, as shown in Fig.(2). By analyzing the structure of CLIP, we found that, after changing the tokenizer, as long as the embedding layers

$f_t^{\mathbf{E}}(\cdot)$ and $f_t^{\mathbf{T}}(\cdot)$ can achieve good alignment in different language, the parameters of the subsequent Transformer layers can be inherited directly, allowing the final output features to align well as well. To achieve this, in the first stage, we use the cross-language alignment as supervision to train the embedding layer of $f_t^{\mathbf{T}}(\cdot)$, which is randomly initialize. Due to the differences in token counts between Chinese and English for the same concepts, the embedding layers may not align perfectly. Therefore, we train the first half of the text encoder's parameters to enable the fusion of varying numbers of tokens. However, aligning only the textual encoders may result in a greater gap between the Chinese encoder and the original image encoder. Therefore, in the final stage, we use the SigLIP loss to directly align the Chinese encoder $f_t^{\mathbf{T}}(\cdot)$ with the image encoder $f_I(\cdot)$. In addition, To enable the model to support both Chinese and English, we implemented a random sampler that randomly selects either Chinese or English captions from the triplet data for training during the three stages.

To obtain high-quality English-Chinese caption and image triplets, we designed a Translation Filtering Network (TFN), which uses the Chinese and English encoders obtained from the second stage to encode the translated Chinese and English captions, respectively, and calculates their similarity to filter out low-similarity caption pairs.

### 3.3 CROSS-LANGUAGE ALIGNMENT

To reduce training costs, we first align the text side in stages 1 and 2, which avoids the need to download hundreds of terabytes of image data and accelerates the training process. To achieve this, we propose cross-language alignment, which treats different languages as distinct modalities within CLIP. This approach facilitates contrastive learning to align representations across various languages.

#### 3.3.1 EMBEDDING LAYER ALIGNMENT

As mentioned above, breaking down the CLIP model reveals that one efficient method to convert an English CLIP into a target language CLIP is to directly replace the word embeddings and tokenizer. Once replaced, if the word embeddings for the same concept in both English and the target language are identical, the representations obtained after passing through the subsequent shared Transformer layers will also be the same. Then, perfect alignment can be achieved.

An intuitive way to replace the embedding layer is to retrieve the corresponding concepts in Chinese and English, and directly use the English embedding to replace the Chinese embedding of the same concept. However, this method cannot exhaust all concepts, which will cause many Chinese embeddings to be vacant and cause instability in training.

In this paper, we propose using cross-language alignment as a supervisory signal, freezing the parameters of the Transformer layers while only training the parameters of the embedding layer so that the embedding layers can be aligned effectively. Similar to the training method used in CLIP for image-text matching, cross-language alignment aims to bring matching Chinese and English captions closer together while pushing non-matching captions further apart.

Formally, let's denote the English caption as $T_i^{\mathbf{E}}$ and its tokens after the original tokenizer as $\text{Tok}_i^{\mathbf{E}}$. Similarly, let the Chinese caption be denoted as $T_i^{\mathbf{T}}$ and its tokens after the Chinese tokenizer be represented as $\text{Tok}_i^{\mathbf{T}}$. The features for the English and Chinese captions, i.e. $\mathbf{z}_i^{T^{\mathbf{E}}}$ and $\mathbf{z}_i^{T^{\mathbf{T}}}$ can be extracted as follows:

$$\mathbf{z}_i^{T^{\mathbf{E}}} = \phi\Big(\mathcal{T}^{\mathbf{E}}\big(\mathcal{W}^{\mathbf{E}}(\text{Tok}_i^{\mathbf{E}})\big)\Big) \tag{4}$$

$$\mathbf{z}_i^{T^{\mathbf{T}}} = \phi\Big(\mathcal{T}^{\mathbf{E}}\big(\widehat{\mathcal{W}}^{\mathbf{T}}(\text{Tok}_i^{\mathbf{T}})\big)\Big) \tag{5}$$

where $\mathcal{W}^{\mathbf{E}}$ and $\widehat{\mathcal{W}}^{\mathbf{T}}$ represent the embedding layers of the English and Chinese encoders, respectively, and $\widehat{\mathcal{W}}$ with superscript indicates that these layers are trainable. Here, the embedding layer contains both word embedding and position embedding, since the word order in Chinese and English may be different. $\mathcal{T}^{\mathbf{E}}$ is the Transformer layers of English encoder, which is also used by Chinese encoder. $\phi(\cdot)$ is the $L_2$ normalization.

After obtaining the English caption feature $\mathbf{z}_i^{T^{\mathbf{E}}}$ and Chinese caption feature $\mathbf{z}_i^{T^{\mathbf{T}}}$, we employ the SigLIP loss (Zhai et al., 2023) function to minimize the distance between matching captions while

maximizing the distance between non-matching ones, where the form of SigLIP loss can be expressed as follows:

$$\mathcal{L}_{siglip} = -\frac{1}{N_B} \sum_{i=1}^{N_B} \sum_{j=1}^{N_B} \log \frac{1}{1 + e^{z_{ij} \cdot (-\tau \cdot \mathbf{z}_i^{T^{\mathbf{E}}} \cdot \mathbf{z}_i^{T^{\mathbf{T}}} + b)}} \quad (6)$$

where $N_B$ is the batch size, $z_{ij}$ is the label for given English and Chinese caption, which equals 1 if they are paired and $-1$ otherwise. $\tau$ is a learnable temperature parameter, and $b$ is a learnable bias. By optimizing the Eq.(6), the Chinese embedding layer can be adequately trained and aligned as closely as possible with the English embedding layer.

### 3.3.2 Token Fusion

The alignment process mentioned above only aligns the embedding layer, while the subsequent Transformer layers directly use the English version $\mathcal{T}^{\mathbf{E}}$, making it difficult to achieve perfect alignment in practice, since the number of tokens representing the same concept or caption can differ across languages. For example, the word "a" in English might be tokenized into a single token, whereas in Chinese, it could be split into two tokens. This discrepancy leads to some deviations in the above alignment.

In the second stage of training, we aim to address the misalignment caused by differences in tokenization between the English and Chinese text encoders. Instead of manually fusing token embeddings, we introduce a novel approach where only the first half of the Chinese encoder's parameters are fine-tuned. Thus, the features for English and Chinese captions are extracted as:

$$\mathbf{z}_i^{T^{\mathbf{E}}} = \phi\Big(\mathcal{T}^{\mathbf{E}}\big(\mathcal{W}^{\mathbf{E}}(\mathrm{Tok}_i^{\mathbf{E}})\big)\Big) \quad (7)$$

$$\mathbf{z}_i^{T^{\mathbf{T}}} = \phi\Big(\widehat{\mathcal{T}}^{\mathbf{T}}\big(\widehat{\mathcal{W}}^{\mathbf{T}}(\mathrm{Tok}_i^{\mathbf{T}})\big)\Big) \quad (8)$$

The difference between the above process and stage 1 is that the parameters of the Transformer layer for Chinese encoder change from frozen $\mathcal{T}^{\mathbf{E}}$ to trainable $\widehat{\mathcal{T}}^{\mathbf{T}}$, where $\widehat{\mathcal{T}}^{\mathbf{T}}$ is initialized with $\mathcal{T}^{\mathbf{E}}$. Here, only the first half of the parameters of $\widehat{\mathcal{T}}^{\mathbf{T}}$ are trainable, the second half of the parameters are frozen. The loss function for optimization is the same as Eq.(6) in stage 1.

This fine-tuning allows the model to implicitly learn how to combine token embeddings representing the same concept in the Chinese. At the same time, due to the freezing of the second half of the parameters, the memory usage and time of training are significantly reduced.

### 3.4 Cross-modal Alignment

After the first two stages of training, the English text encoder and the Chinese text encoder have achieved an initial alignment. However, due to the fact that the training loss in the second phase mentioned above never reaches zero, there remains a persistent gap between the English text encoder and the Chinese text encoder. Similarly, a gap exists between the English text encoder and the image encoder, which further increases the gap between the Chinese text encoder and the image encoder. Therefore, it is necessary to directly align the Chinese text encoder with the image encoder.

Similarly, we also use SigLIP loss (Zhai et al., 2023) for alignment and optimization in the third stage. Here, the feature extraction process is as follows:

$$\mathbf{z}_i^I = \phi\big(f_I(I_i)\big) \quad (9)$$

$$\mathbf{z}_i^{T^{\mathbf{T}}} = \phi\Big(\widehat{\mathcal{T}}^{\mathbf{T}}\big(\widehat{\mathcal{W}}^{\mathbf{T}}(\mathrm{Tok}_i^{\mathbf{T}})\big)\Big) \quad (10)$$

where we unfreeze all parameters in the Chinese Transformer $\widehat{\mathcal{T}}^{\mathbf{T}}$ for training in the third stage. With image feature $\mathbf{z}_i^I$ and text feature $\mathbf{z}_i^{T^{\mathbf{T}}}$, the SigLIP loss is formalized as:

$$\mathcal{L}_{siglip} = -\frac{1}{N_B} \sum_{i=1}^{N_B} \sum_{j=1}^{N_B} \log \frac{1}{1 + e^{z_{ij} \cdot (-\tau \cdot \mathbf{z}_i^I \cdot \mathbf{z}_i^{T^{\mathbf{T}}} + b)}} \quad (11)$$

where $z_{ij}$ is the label for given image and text input. $\tau$ is a learnable temperature parameter, and $b$ is a learnable bias.

### 3.5 MULTI-LANGUAGE SUPPORT

To enable the model to support both Chinese and English, we implemented a random sampler that randomly selects either Chinese or English captions from the triplet data for training during the three stages. Specifically, during the training phases, there is a 50% chance that a Chinese caption will be replaced by its corresponding English caption.

### 3.6 TRANSLATION FILTERING NETWORK

To complete the training process described above, we require two main types of data: 1) English–Chinese caption pairs for the first and second stages of training. 2) English–Chinese caption and image triples for the third phase of training. Since most open source image and text datasets are in English, we used Qwen 1.5 (Bai et al., 2023) to translate the open source dataset DFN2B (Fang et al., 2024). We translated a total of 1.5 billion English captions into Chinese.

However, since the output of LLM is unstable, that is, it is easy to output a lot of irrelevant nonsense or refuse to translate, we designed a Translation Filtering Network (TFN) to filter out low-quality translations. Due to the above stage 2, we can obtained a very well aligned Chinese Encoder and English Encoder, which can well measure the similarity between Chinese caption and English caption. Therefore, our Translation Filtering Network consists of the two Encoders, and its filtering process is as follows:

1. Eq.(7) and Eq.(8) are used to calculate the features of Chinese caption $T_i^{\mathbf{E}}$ and English caption $T_i^{\mathbf{T}}$ respectively, where the obtained feature is denoted as $\mathbf{z}_i^{T^{\mathbf{E}}}$ and $\mathbf{z}_i^{T^{\mathbf{T}}}$.

2. The similarity between $\mathbf{z}_i^{T^{\mathbf{E}}}$ and $\mathbf{z}_i^{T^{\mathbf{T}}}$ are measured, and the samples that are higher than the threshold are kept. The resulting high-quality data pairs can be expressed as $\Omega = \{(T_i^{\mathbf{E}}, T_i^{\mathbf{T}})_{i=1}^{\mathcal{N}} | \mathbf{z}_i^{T^{\mathbf{E}}} \cdot \mathbf{z}_i^{T^{\mathbf{T}}} \geq \Theta\}$, where $\Theta$ is predefined threshold, $\mathcal{N}$ is the number of translated captions.

Through the above TFN, we filter out about 0.3 billion translations and can get high-quality 1.2 billion Chinese-English caption pair $\Omega$, which will be used to retrain the above stages 1 and 2.

## 4 EXPERIMENTS

### 4.1 MODEL ZOO AND IMPLEMENTATION DETAILS

#### 4.1.1 MODEL ZOO

The YouCLIP model includes three different sizes: Base, Large, and Huge, and it supports various input image resolutions such as 224x224, 256x256, 384x384, and 512x512. Additionally, we have trained pure Chinese versions and bilingual (Chinese and English) versions for each size. Detailed information on the model's parameters, computational complexity can be found in Tab.1, where models ending with "DL" indicate versions that support double language (Chinese and English), while the others are pure Chinese models. For the pure Chinese models, we removed the random sampler and used only Chinese for training in all three stages. The "token len" in the table refers to the maximum token length for the text input, and The "Dim" represents the output feature dimension of the model.

#### 4.1.2 PRETRAINING DETAILS

All versions of the YouCLIP model were trained in parallel using 48 V100 GPUs. The overall framework is based on the open-clip (Ilharco et al., 2021) architecture, and all models underwent fine-tuning from Google's SigLIP (Zhai et al., 2023) model. Throughout the training process, the image side remained frozen, while the text side's learning rate gradually decreased over three stages: 1e-4 in the first stage, 1e-5 in the second, and 2e-6 in the third. Additionally, all stages included a warm-up set to 5,000 iterations. The Translation Filtering Network utilizes the YouCLIP-Base for filtering. All YouCLIP models and code will be open-sourced.

| Model Zoo | Params | GFLOPs | Dim | Language | Resolution | Token Len |
|---|---|---|---|---|---|---|
| YouCLIP-Base | 209M | 47.1 | 768 | Chinese only | 224 | 77 |
| YouCLIP-Base-DL | 209M | 47.1 | 768 | Chinese + English | 224 | 77 |
| YouCLIP-Base-512 | 210M | 190.6 | 768 | Chinese only | 512 | 77 |
| YouCLIP-Base-512-DL | 210M | 190.6 | 768 | Chinese + English | 512 | 77 |
| YouCLIP-Large | 660M | 194.8 | 1024 | Chinese only | 256 | 64 |
| YouCLIP-Large-DL | 660M | 194.8 | 1024 | Chinese + English | 256 | 64 |
| YouCLIP-Huge | 1016M | 656.8 | 1152 | Chinese only | 384 | 64 |
| YouCLIP-Huge-DL | 1016M | 656.8 | 1152 | Chinese + English | 384 | 64 |

Table 1: Model Zoo of YouCLIP. YouCLIP includes versions with different parameter amounts, different resolutions, and different language support.

## 4.2 EVALUATION DATASET AND METRICS

We evaluate our model on the following tasks: Chinese cross-modal retrieval, Chinese zero-shot classification and English zero-shot classification. For Chinese cross-modal retrieval, we evaluate on COCO-CN(Li et al., 2019), Flickr30K-CN(Lan et al., 2017) and MUGE. For Chinese zero-shot classification, we evaluate on ELEVATER(Li et al., 2022) benchmark. For English zero-shot classification, we evaluate on ImageNet-1K. Details of datasets can be found in Appendix.A.

For retrieval tasks, we evaluate using Recall@1, Recall@5, and Recall@10 metrics. For zero-shot classification tasks, we use the default metrics corresponding to the dataset, such as Accuracy, Mean-per-class, and 11-point mAP.

## 4.3 EVALUATION OF CHINESE CROSS-MODAL RETRIEVAL

We evaluate on three Chinese cross-modal retrieval datasets, namely COCO-CN, Flickr30K-CN and MUGE. Tab.2 presents the results. For COCO-CN, we observe that the YouCLIP series significantly outperforms previous SOTA models. For the Base version, YouCLIP-Base enhances the Recall@1 metric for image to text from 57.0 to 70.5, marking an improvement of 13.5 points. Similarly, the text to image metric increases by 5.2 points. This trend continues for the Large and Huge models as well; the Large model raises the previous text to image score from 64.7 to 70.7, while the Huge model improves the image to text score from 63 to 73.1, achieving a total increase of 10.1 points.

The same trend is observed for Flickr30K-CN and MUGE. For the results of Flickr30K-CN, the Recall@1 metric for text to image for the Base, Large, and Huge versions improved by 9.1, 8.7, and 10.9 points, respectively, demonstrating significant enhancement and validating the effectiveness of our method. For MUGE, the YouCLIP models of different sizes achieved improvements of 9.8, 5.7, and 5.0 points compared to CN-CLIP in the text to image task. This strong performance further confirms the efficacy of the YouCLIP approach.

## 4.4 EVALUATION OF CHINESE AND ENGLISH ZERO-SHOT CLASSIFICATION

Tab.3 shows the Chinese and English zero-shot classification performance of large size model. The table is divided into two parts: the upper section displays the performance of the original English model on English datasets, while the lower section shows the performance of the Chinese model on translated Chinese datasets. Additionally, the far-right column presents the results for ImageNet-1K, an English dataset. From the table, it is evident that YouCLIP not only achieves high performance on Chinese benchmarks but also maintains strong performance in English. Among the ten Chinese classification datasets listed, YouCLIP achieves the highest performance in six of them. Furthermore, YouCLIP attains a top-1 accuracy of **76.9** on ImageNet-1K, demonstrating its robust capabilities in English as well.

## 4.5 ABLATION STUDY

### 4.5.1 THE IMPACT OF DIFFERENT TRAINING STAGES

Here, we investigate the impact of different training stages on the final performance to explore the importance and indispensability of each stage. Tab.4 presents the Recall@1 retrieval metrics on

| Methods | COCO-CN | | | | Flickr30K-CN | | | | MUGE | | | |
| | Image-to-Text | | Text-to-Image | | Image-to-Text | | Text-to-Image | | Image-to-Text | | Text-to-Image | |
| | R@1 | R@5 | R@1 | R@5 | R@1 | R@5 | R@1 | R@5 | R@1 | R@5 | R@1 | R@5 |
|---|---|---|---|---|---|---|---|---|---|---|---|---|
| Wukong-Base | 48.3 | 77.8 | 49.2 | 79.4 | 66.2 | 88.7 | 45.7 | 73.8 | – | – | 33.4 | 59.3 |
| CN-CLIP-Base | 57.0 | 84.1 | 62.2 | 86.6 | 74.6 | 93.5 | 62.7 | 86.9 | – | – | 52.1 | 76.7 |
| **YouCLIP-Base-DL** | 70.1 | 92.1 | 68.2 | 90.1 | 88.9 | 98.8 | 71.0 | 91.5 | 48.4 | 77.0 | 57.0 | 81.3 |
| **YouCLIP-Base** | **70.5** | **92.4** | 67.4 | **90.7** | 88.7 | 98.9 | 71.8 | 92.1 | **53.2** | **81.7** | 61.9 | 85.3 |
| Wukong-Large | 55.2 | 81.0 | 53.4 | 80.2 | 76.1 | 94.8 | 51.7 | 78.9 | – | – | 42.7 | 69.0 |
| CN-CLIP-Large | 60.4 | 84.2 | 64.0 | 89.2 | 80.2 | 96.6 | 68.0 | 89.7 | – | – | 56.3 | 79.8 |
| CN-CLIP-Large-336 | 63.4 | 87.2 | 64.7 | 89.6 | 83.3 | 97.2 | 69.0 | 90.7 | – | – | 59.0 | 81.4 |
| **YouCLIP-Base-512-DL** | 70.7 | 92.8 | 69.1 | 91.0 | 92.8 | 98.9 | 77.7 | 94.2 | 53.8 | 82.8 | 61.8 | 85.7 |
| **YouCLIP-Base-512** | 72.5 | 92.8 | 69.0 | **91.4** | 93.5 | 99.0 | 77.3 | 94.2 | 52.4 | 81.3 | 61.0 | 84.4 |
| **YouCLIP-Large-DL** | 73.1 | 93.3 | 70.1 | 90.6 | 90.8 | 99.3 | 76.8 | 93.4 | 54.1 | 82.5 | 62.3 | 85.2 |
| **YouCLIP-Large** | **73.1** | **93.4** | 70.7 | 90.9 | 92.1 | **99.4** | 77.2 | 93.6 | **56.3** | **84.6** | 64.7 | 87.1 |
| CN-CLIP-Huge | 63.0 | 86.6 | 69.2 | 89.9 | 81.6 | 97.5 | 71.2 | 91.4 | – | – | 63.0 | 84.1 |
| **YouCLIP-Huge-DL** | 72.0 | 93.1 | 71.2 | 92.5 | 95.2 | 99.5 | 81.3 | 96.1 | 58.1 | 86.2 | 66.5 | 88.9 |
| **YouCLIP-Huge** | **73.1** | **94.0** | 71.9 | 92.9 | 95.9 | 99.7 | 82.1 | 96.3 | 58.6 | 86.7 | 68.0 | 89.1 |

Table 2: Zero-shot retrieval results on COCO-CN, Flickr30K-CN, and MUGE datasets.

| Model | CIFAR10 | CIFAR100 | DTD | EuroSAT | FER | FGVC | KITTI | MNIST | PC | VOC | ImageNet(EN) |
|---|---|---|---|---|---|---|---|---|---|---|---|
| *Original benchmark* | | | | | | | | | | | |
| DeCLIP | 90.9 | 66.8 | 44.9 | 39.9 | 23.3 | 9.0 | 39.7 | 13.6 | 55.3 | 80.6 | 73.7 |
| ALIGN | **94.9** | 76.8 | **66.1** | 52.1 | **50.8** | 25.0 | **41.2** | 74.0 | 55.2 | 83.0 | **76.4** |
| OpenCLIP | 93.5 | 76.2 | 56.4 | 53.7 | 50.3 | 20.8 | 28.8 | 70.9 | 50.5 | 82.3 | |
| CLIP | **94.9** | **77.0** | 56.0 | **63.0** | 48.3 | **33.3** | 11.5 | **79.0** | **62.3** | **84.0** | 76.2 |
| *Chinese benchmark* | | | | | | | | | | | |
| BriVL | 72.3 | 35.9 | 18.8 | 25.5 | - | - | - | - | - | - | 24.3 |
| Wukong | 95.4 | 77.1 | 40.9 | 50.3 | - | - | - | - | - | - | 55.0 |
| CN-CLIP | 96.0 | 79.7 | 51.2 | 52.0 | **55.1** | 26.2 | **49.9** | 79.4 | **63.5** | **84.9** | 59.6 |
| YouCLIP | **96.5** | **82.0** | **61.0** | 53.3 | 51.6 | **57.8** | 36.1 | **88.9** | 50.0 | 22.8 | **76.9** |

Table 3: Experimental results of the Chinese and English zero-shot image classification performance on ELEVATER and ImageNet-1K. All models are Large size models.

COCO-CN text to image for models of varying sizes. The implementation details are as follows: 1) For models without Stage 1, the embedding layer is randomly initialized for subsequent training. 2) To ensure fairness, all training configurations maintain the same total number of epochs.

From Tab.4, we can observe the following: 1) **Training solely with Stage 1 already achieves strong performance**, surpassing CN-CLIP. This demonstrates that our approach to training the embedding layer is both effective and reasonable. 2) Training only with Stage 2 results in poorer performance, likely due to the instability caused by the random initialization of the embedding layer. The embedding layer training in Stage 1 provides stable improvements for both Stage 2 and Stage 3. 3) Stage 2 training, based on Stage 1, consistently boosts performance by approximately 1%. 4) The combination of all three stages achieves the highest performance.

### 4.5.2 CHANGES IN MODEL PERFORMANCE DURING THE THREE TRAINING PHASES

Here, we explore how the model's performance evolves across different training stages to examine the impact of each stage. Fig.3 illustrates the performance metrics of models of various sizes across three retrieval datasets. From this, we can observe that the model experiences a rapid performance increase during the first stage, primarily due to the fast alignment and convergence achieved by training the embedding component separately. In the second and third stages, the model's performance continues to improve steadily.

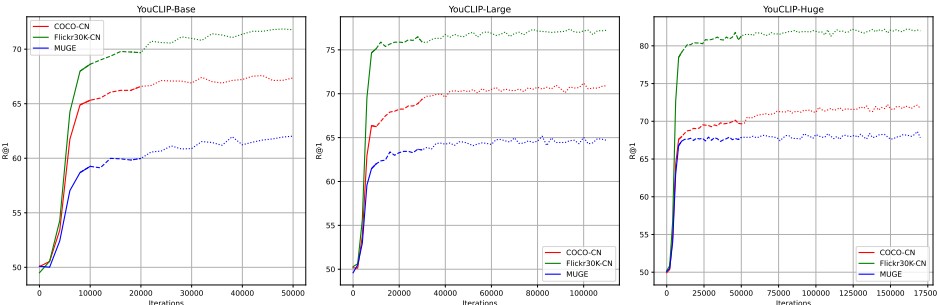

Figure 3: Changes in model performance during the three training phases. Different line styles represent different stages: solid line-1st, dashed line-2nd, dotted line-3st.

| Stage 1 | Stage 2 | Stage 3 | YouCLIP-Base | YouCLIP-Large | YouCLIP-Huge |
|---------|---------|---------|--------------|---------------|--------------|
| ✓ | | | 65.1 | 66.9 | 68.2 |
| | ✓ | | 59.0 | 61.3 | 65.0 |
| | | ✓ | 64.5 | 66.3 | 69.7 |
| ✓ | ✓ | | 66.4 | 68.9 | 69.9 |
| ✓ | | ✓ | 66.8 | 69.3 | 70.6 |
| ✓ | ✓ | ✓ | **67.4** | **70.7** | **71.9** |

Table 4: Ablation experiments on the effects of different stages

| Filter Ratio | Number of Reserved | YouCLIP-Base | YouCLIP-Large | YouCLIP-Huge |
|--------------|--------------------|--------------|--------------|--------------|
| 0% | 1.5B | 65.9 | 70.3 | 71.2 |
| 10% | 1.35B | 66.8 | 70.5 | 71.8 |
| 20% | 1.2B | **67.4** | **70.7** | **71.9** |
| 30% | 1.05B | 67.1 | 70.6 | 71.5 |
| 50% | 0.75B | 67.0 | 70.5 | 71.3 |

Table 5: Ablation on Translation Filtering Network

### 4.6 EFFECTS OF TRANSLATION FILTERING NETWORK

Here, we explore the impact of the Translation Filtering Network (TFN) on model performance. We set different filtering ratios and test the performance of the Base, Large, and Huge versions of the model on the COCO-CN dataset using the text-to-image Recall@1 metric. The results are shown in Tab.5. From the results, we can observe the following: 1) The TFN generally leads to performance improvements compared to using the raw data directly. 2) The performance boost from TFN is much more significant for the Base model than for the Huge model, which may be due to the fact that smaller models are more sensitive to data quality.

## 5 CONCLUSION

Currently, most advanced CLIP models are developed primarily for English, making it difficult for non-English-speaking countries to apply these models effectively. To address this, this paper proposes a simple and efficient three-stage fine-tuning approach to convert the most powerful English CLIP model (using the SigLIP model in this study) into a Chinese CLIP model, named YouCLIP. In these three stages of training, the first stage focuses on aligning the embedding layer, followed by token fusion in the second stage, and finally contrastive learning fine-tuning in the third stage. This three-stage process enables YouCLIP to become the most powerful Chinese CLIP model. Not only does it achieve the highest performance across all Chinese CLIP benchmarks, but it also retains strong English capabilities.

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

## A    DETAILS FOR DATASETS

The following is a detailed description of the datasets used for our evaluation.

- COCO-CN for Chinese cross-modal retrieval: COCO-CN(Li et al., 2019) dataset is an extension of the COCO dataset specifically tailored for tasks involving the Chinese language. It features 20,342 images, complemented by 27,218 Chinese sentences and 70,993 tags.

- Flickr30K-CN for Chinese cross-modal retrieval: Flickr30K-CN(Lan et al., 2017) is a Chinese image annotation dataset that serves as an extension of the Flickr30K dataset. It translates or rewrites the original English descriptions into Chinese, providing multiple Chinese descriptions for each image.

- MUGE for Chinese cross-modal retrieval: MUGE is a large-scale Chinese multimodal evaluation benchmark that includes a vast collection of paired image and text data from the E-commerce platform.

- ELEVATER for Chinese zero-shot classification: ELEVATER(Li et al., 2022) is a benchmark dataset and evaluation platform designed to assess the performance of visual and language models on multimodal tasks. The benchmark includes 20 image classification datasets. Some of these focus on fine-grained classification, such as aircraft models, flower species, and pet breeds. Others require more complex reasoning, like determining whether an emoji conveys a negative sentiment or analyzing the spatial relationships between people and cars.

- ImageNet-1K for English zero-shot classification: ImageNet-1K(Deng et al., 2009) is a widely-used image recognition dataset containing 1,000 categories that span a broad range of object types, including animals, everyday items, vehicles, plants, and more. This diversity makes it ideal for training models capable of recognizing a wide variety of objects across different domains.

