# OpenReview forum: "YouCLIP: Advancing Multilingual Cross-Modal Learning with Efficient Training."
_ICLR.cc/2025/Conference — Submitted to ICLR 2025_

### Official Review · Reviewer_hnS7 · 2024-11-01

**Soundness:** 2
**Presentation:** 2
**Contribution:** 2
**Rating:** 3
**Confidence:** 4

**Summary:**

The paper presents a framework to train a non-English CLIP model from English CLIP model. The authors propose three stages of training: 1) Embedding layer alignment; 2) the first half of the Chinese encoder's parameters training; 3)Aligning Chinese encoder with image encoder. The paper instroduces a Translation Filtering Network to filter out the low-quality translated data.

**Strengths:**

The paper presents a knowledge distillation framework for non-English CLIP model training, allowing for the conversion of the English CLIP model into models for other languages. The method is easy to understand and the research question is interesting.

**Weaknesses:**

The motivation of the paper is to improve the performance of CLIP models for multiple languages and the title contains "multilingual". However, the model only supports two languages. It is unclear how the model performs on other languages.

The technical novelty is limited. It is common to align different languages by training the embedding layer and using non-English image-text pairs to align the non-English text encoder and image encoder.

The design of the method is relatively arbitrary. For example, the paper assumes the primary difference of text encoders lies in the embedding layer. However, the embedding layer is trained in stage 1 while the first half of the text encoder are trained  in stage 2.

The model requires a large amount of data, at the billion level, and lacks a detailed comparison with other methods in terms of computational overhead.

**Questions:**

1. The paper uses the qwen 1.5 models to translate English data to Chinese. How about the performance on the low-resource non-English languages?

2. Can you provide more detailed ablation studies comparing different architectural choices?

3. What about the performance when the model is a multilingual model that supports multiple languages?

---

### Official Review · Reviewer_ZtgJ · 2024-11-03

**Soundness:** 3
**Presentation:** 2
**Contribution:** 3
**Rating:** 5
**Confidence:** 4

**Summary:**

This paper tackles a problem of adapting the existing CLIP model that is trained in English into another language such as Chinese. The authors proposed a three-stage training method: 1) the first stage focusing on the embedding layer alignment, 2) token fusion and 3) contrastive learning fine-tuning. Experimental results show effectiveness of the proposed approach, significantly outperforming previous models across all Chinese benchmarks.

**Strengths:**

- Experimental results are technically sound

**Weaknesses:**

- some experimental settings and/or results are unclear; what about applying the proposed approach into other languages? robustness? Have you ever tried few-shot experiments related to Tables 2 and 3?

**Questions:**

- Figure 1 (Left) needs a scale or values for the results of "Wukong" and "CN-CLIP"
- Tab.2 -> Table 2 for better readability, as I don't see any space issues in the current manuscript
- OpenCLIP's result on ImageNet (EN) might be missing in Table 3?

---

### Official Review · Reviewer_33ov · 2024-11-05

**Soundness:** 3
**Presentation:** 3
**Contribution:** 2
**Rating:** 5
**Confidence:** 4

**Summary:**

This paper proposes a method to adapt the English-Image CLIP model to a Chinese-English-Image model by just training a new language encoder without modifying the Image encoder, with a very little loss on English-Image tasks.  The proposed method use a very limited training resources compared with existing Chinese-Image models which are training from scratch with Chinese-Image datasets.   The resulting model YouCLIP creates new SoTAs in almost all Chinese-Image understanding benchmarks.

The method has three steps, while in step one only the embedings of the new language encoder are trained and in the step 2 and 3 both the embeddings and the half of the transformer layers of the language encoder are trained.  In step 2, the language encoder are trained with Chinese-English aligned data and in the step 3, the Chinese encoder are trained with Chinese-Image or English-Image data randomly.

The English-Chinese-Image data are construsted by translating the English captions into Chinese in an English-Image dataset.  The translation is conducted with a strong LLM QWEN 1.5.  A translation filtering network is proposed is designed to filter out low quality translations.

**Strengths:**

The paper is clearly written and the experiments have well demonstrate the effectiveness of the proposed method.
The paper creates a new SoTA for Chinese version of CLIP by adapting the English CLIP with the help of an LLM: QWEN 1.5.
The proposed method includes three steps is introduced in details.  Compare with the previous work which training the model from the scratch, this method requires much less computing resources.

**Weaknesses:**

I wonder the high performance of YouCLIP not only comes from the proposed methods, but also heavily depends on the translation system.  The paper uses QWEN 1.5 for the translation from English to Chinese, which is much larger than CLIP itself.  So the claim that the proposed requires the least amount of training resources is not really true: it uses an existing LLM which is trained with huge amount of resouces which is much larger than the training of a Chinese CLIP.

The effect of the translation system to produce the triple data should be analysed.  Although the paper analyses the effect of the AFN to the final performance, it is not enough.

More details of the AFN also should be provided.

I further suggest the auther to add a pipeline system as the baseline, which first translation Chinese captions into English with the same translation system, than use the original CLIP system with the English captions and the image.  I am curious if YouCLIP can outperform this pipeline system.

**Questions:**

See weaknesses above.

---

### Meta-Review · Area_Chair_Mu3w · 2024-12-25

**Metareview:**

This paper introduces a three-stage pipeline for adapting an English-only CLIP model into a bilingual Chinese-English-Image model (“YouCLIP”) without modifying the existing image encoder. The resulting model demonstrates competitive or state-of-the-art performance on various Chinese-language image understanding benchmarks, purportedly with reduced training costs compared to training a Chinese CLIP model from scratch.

**Strength**: (1) High Performance on Chinese Benchmarks (2) Efficient Reuse of English CLIP

**Weakness** (1) Heavy Reliance on a Large LLM. As mentioned by reviewers, the claim of “reduced training resources” is questionable as the strong translator (QWen 1.5) has already cost substantial cost; (2) limited technical novelty and scope; (3) Insufficient analysis and ablations.

**Decision**: Although YouCLIP achieves impressive Chinese-English performance and offers a conceptually straightforward approach for reusing pre-trained CLIP image encoders, reviewers are not uniformly convinced of the paper’s broader contributions or resource-efficiency claims.  Given these limitations—particularly the questions around novelty, limited language scope, I am leaning toward reject.

**Additional Comments On Reviewer Discussion:**

No discussion happened since the author chose not to respond in the rebuttal phase.

---

### Decision · Program_Chairs · 2025-01-22

Reject